# A Review of the Most Recent Clinical and Neuropathological Criteria for Chronic Traumatic Encephalopathy

**DOI:** 10.3390/healthcare11121689

**Published:** 2023-06-08

**Authors:** Ioannis Mavroudis, Ioana-Miruna Balmus, Alin Ciobica, Alina-Costina Luca, Dragos Lucian Gorgan, Irina Dobrin, Irina Luciana Gurzu

**Affiliations:** 1Department of Neurology, Leeds Teaching Hospitals NHS Trust and Leeds University, Leeds LS9 7TF, UK; 2Department of Exact Sciences and Natural Sciences, Institute of Interdisciplinary Research, Alexandru Ioan Cuza University of Iasi, Alexandru Lapusneanu Street, No. 26, 700057 Iasi, Romania; 3Department of Biology, Faculty of Biology, Alexandru Ioan Cuza University of Iasi, B dul Carol I, No. 11, 700506 Iasi, Romania; 4Faculty of Medicine, “Grigore T. Popa” University of Medicine and Pharmacy of Iasi, 700115 Iasi, Romania; 5Department of Preventive Medicine and Interdisciplinarity, Discipline of Occupational Medicine, Faculty of Medicine, “Grigore T. Popa” University of Medicine and Pharmacy of Iasi, 700115 Iasi, Romania

**Keywords:** chronic traumatic encephalopathy, traumatic encephalopathy syndrome, traumatic brain injury, diagnosis algorithm, brain imaging, neuropathological features

## Abstract

(1) Background: Chronic traumatic encephalopathy (CTE) is a complex pathological condition characterized by neurodegeneration, as a result of repeated head traumas. Currently, the diagnosis of CTE can only be assumed postmortem. Thus, the clinical manifestations associated with CTE are referred to as traumatic encephalopathy syndrome (TES), for which diagnostic multiple sets of criteria can be used. (2) Objectives: In this study, we aimed to present and discuss the limitations of the clinical and neuropathological diagnostic criteria for TES/CTE and to suggest a diagnostic algorithm enabling a more accurate diagnostic procedure. (3) Results: The most common diagnostic criteria for TES/CTE discriminate between possible, probable, and improbable. However, several key variations between the available diagnostic criteria suggest that the diagnosis of CTE can still only be given with postmortem neurophysiological examination. Thus, a TES/CTE diagnosis during life imposes a different level of certainty. Here, we are proposing a comprehensive algorithm of diagnosis criteria for TES/CTE based on the similarities and differences between the previous criteria. (4) Conclusions: The diagnosis of TES/CTE requires a multidisciplinary approach; thorough investigation for other neurodegenerative disorders, systemic illnesses, and/or psychiatric conditions that can account for the symptoms; and also complex investigations of patient history, psychiatric assessment, and blood and cerebrospinal fluid biomarker evaluation.

## 1. Introduction

Chronic traumatic encephalopathy (CTE) was first described in 1928 as the neuropsychiatric sequelae of repetitive head impacts in boxing athletes, for which the term “punch drunk” syndrome was used [1]. Several decades later, in 1954, Brandenburg and Hallervorden [2] published the first neuropathological reports of this condition, while 19 years later, Corsellis and colleagues described pathological findings associated with the condition known as *dementia pugilistica*, following a systematic neuropathological study on former professional and amateur boxers [3].

Although CTE was described almost 100 years ago, it was only less than 20 years ago when Dr. Bennet Omalu began a powerful program to raise awareness of its importance, public burden, and increased impact on affected people’s wellbeing, as well as increased mortality rates, all of which culminated with the premiere of the American biographical sports drama film, “Concussion”, in 2015. It is now understood that this condition is not as uncommon as we thought before and that it can occur in athletes involved in contact sports, such as boxing, martial arts, American football, and rugby, as well as soccer, water polo, and ice hockey [4,5,6,7,8,9,10,11]. Furthermore, it was described in military personnel with a yet unsolved correlation with post-traumatic stress disorder (PTSD) [11,12].

CTE is considered a rare, progressive, and fatal brain disorder that is most often associated with repeated traumatic brain injury (TBI). Despite that, other causes of CTE are not yet described, and the pathophysiology of CTE is yet not very well understood and described. The physiology of the CTE-affected brain is often specific for neurodegeneration and includes brain matter atrophy, white matter perivascular spaces, and, rarely, cerebellar abnormalities [13]. While the pathological profiles of younger TES patients are often characterized by behavioural and mood impairments, it was shown that older patients often exhibit symptoms similar to the hallmarks of neurodegeneration as seen in Alzheimer’s disease [14,15].

The neurodegenerative processes in CTE were described as the accumulation of neurofibrillary tangles (NFT) as a result of tau protein hyperphosphorylation, amyloid beta depositions, and immunoreactive inclusion bodies [13,14]. The mechanisms through which repetitive TBIs are leading to neurodegenerative changes in the brain are currently proposed to include chronic inflammation, amyloid cascade activation, axonal protein degeneration, and neuronal homeostasis disbalance [15]. However, these mechanisms are not completely described and understood within the pathophysiological changes in TES/CTE. 

The clinical implications of CTE were found to include neuropsychiatric symptoms, such as behavioural changes, suicidality, cognitive dysfunction, motor disturbance, and emotional dysregulation [16]. However, currently, the diagnosis of CTE can only be assumed postmortem. Thus, the clinical manifestations associated with CTE are referred to as traumatic encephalopathy syndrome (TES), according to multiple sets of diagnostic criteria. 

Therefore, in this review, we aimed to present and discuss the limitations of the clinical and neuropathological diagnostic criteria for TES/CTE and to suggest a novel diagnostic algorithm enabling a more accurate diagnostic procedure.

## 2. Clinical Diagnostic Criteria and Diagnostic Algorithm

According to the recent guidelines (Table 1), TES can be classified based on different diagnostic criteria, proposed by different groups [16,17,18,19], the most common of which discriminate between possible, probable, and improbable TES [20]. Despite the slight differences in terms, all the previously proposed diagnostic criteria include exposure to significant and repetitive head impacts, the progressive character of the symptoms, and their persistence for at least one year. However, a valid diagnostic requires the presence of behavioural/cognitive impairment, motor symptoms, and neuropathological confirmation [21,22]. 

Montenigro et al. suggested that TES/CTE diagnosis procedure requires exposure to repetitive head trauma events, one core clinical feature (such as cognitive, behavioural, or mood disturbance) alongside two supportive features (including headaches, motor signs, impulsivity, anxiety, apathy, paranoia, suicidality, and progressive course, or delayed onset), and symptom persistence for longer than one year in the absence of another condition that could account for the symptoms [20]. 

On the other hand, Reams et al. proposed that the diagnosis should be confirmed if the patient has symptoms that persisted for longer than two years (and no other neurologic disorder can explain those symptoms), history of exposure to traumatic events with concussive and/or sub-concussive head traumas, confirmed progressive course, late symptom onset, and cognitive decline confirmed by neuropsychological testing [19]. These diagnostic criteria also provide a classification for TES/CTE according to the predominant symptomatology (behavioural or mood variant, cognitive variant, mixed variant, and dementia variant) and based on symptom progression (progressive, stable, and unknown or inconsistent type) [19].

A panel of 20 experts in the fields of neurology, neuropsychology, psychiatry, neurosurgery, rehabilitation, and physical medicine affiliated with different academic institutions reviewed the evidence from all the reported cases of definite TES/CTE and published the diagnostic criteria known as The National Institute of Neurological Disorders and Stroke Consensus Diagnostic Criteria for TES [21]. These criteria require (1) substantial exposure to repetitive head trauma events from contact sports, military service, or other causes; (2) core clinical features of cognitive impairment in episodic memory and/or executive functioning, and/or neurobehavioural dysregulation; and (3) progressive course of the condition, (4) in the absence of any other neurologic, psychiatric, or medical condition. However, biomarkers were not included, as the technological and scientific developments in the field are not sufficiently mature, according to NINDS Consensus [21]. 

Furthermore, Jordan et al. suggested that probable TES/CTE should be diagnosed if two or more of the following are present: cognitive and/or behavioural impairment, cerebellar dysfunction, and pyramidal or extrapyramidal tract symptoms. Meanwhile, Katz et al. suggested that the diagnosis should be supported by abnormal findings from positron emission tomography, single-emission tomography, structural magnetic resonance imaging, or diffusion-tensor imaging [20,21]. 

Thus, it seems that cognitive impairment is one of the most common clinical symptoms in patients diagnosed with TES/CTE, being reported in more than 60% of the cases, while the behavioural features, such as violent, impulsive, or explosive behaviour, socially inappropriate behaviour, aggression, rage, short fuse, and lack of behavioural control, are identified in more than 40% of the cases. Mood changes, anxiety, and paranoid delusions are also frequently reported, and more than 30% of TES/CTE patients are diagnosed with depression, anxiety, hopelessness, and apathy. A progressive cognitive decline is seen in more than 95% of cases, whereas more than 50% of TES/CTE patients may develop problems with balance or gait, 23% develop dysarthria, and 28% develop parkinsonism [21]. Furthermore, despite the general understanding that CTE could be mainly an issue of repetitive head impacts over time, it was recently suggested that the behavioural response to TBI could be a major predisposing factor in CTE development. For instance, Ganau et al. [22] proposed that agitation and delirium seen in TBI patients could be significant risk factors that predict long-term changes within the patients’ neurophysiology and thus possibly promote CTE onset.

This seems to indicate that structural and functional brain damage can contribute to creating the basis for CTE in the long term.

## 3. Diagnostic Algorithm

### 3.1. Symptoms Recognition and Evaluation

Patients usually seek medical assistance for non-specific symptoms, such as headaches, dizziness, fatigue, physical exercise intolerance, light and noise sensitivity, mild memory impairment, impulsive behaviour, and/or mood changes including depression and apathy. The first symptoms are either noticed by family and friends or by the patients themselves. In this case, good clinical history, detailed physical examination, and diagnostic tests are critical for a clean diagnosis process. While evaluating a patient for TES, the main questions that a medical doctor needs to consider are as follows: (1) Is there any history of substantial exposure to repetitive head injuries? (2) Is there any history of moderate or severe TBI? (3) How long have the symptoms persisted? (4) Are the symptoms progressive?

### 3.2. Exposure to Head Trauma

The history of substantial exposure to repetitive head impacts, or the history of moderate or severe TBI, is mandatory in all the proposed sets of diagnostic criteria. However, clinical symptoms of concussion or post-concussion syndrome are not required in the mandatory condition definition. Thus, in order to further clarify the history to which the criterion applies, some specific conditions have been proposed, including the following:1.At least 5 years of active and organized participation in high-exposure contact sports, such as boxing, wrestling and martial arts, cage fighting, American football and rugby, ice hockey, or soccer.2.Vocational activities that predispose a person to repeated head impacts: military service, leading to exposure to blasts and other explosions or multiple blows to the head; policing or other task force work, leading to exposure due to breaching locked doors and other barriers as a first responder; and other vocational activities, which may be exposed to various injuries involving multiple head impacts.3.Domestic violence and different types of abuse survivors.4.Individuals with head-banging behaviour, such as patients diagnosed with neuropsychiatric diseases characterized by repetitive self-harming episodes [23,24].

### 3.3. Core Clinical Features

When the first diagnostic criterion is met, the patient should be evaluated for the core clinical features. Cognitive impairment or emotional/behavioural dysregulation are required to meet this criterion. 

Cognitive impairment should include self-reported symptoms or symptoms recognized by the clinician. Significant decline from baseline functioning, deficits in episodic memory and/or executive functioning, and substantially impaired performance on formal neuropsychological testing should account for a positive diagnostic criterion. Similarly, neurobehavioral dysregulation could require self-reported or clinically confirmed poor regulation or control of emotions and/or behaviour. 

Furthermore, a very important characteristic of TES/CTE’s core clinical features is the progressive course of the symptoms. In this context, the positive diagnostic criterion requires that the symptoms lasted for at least one year in the absence of continued exposure to repetitive head impacts, while the progression should be supported by serial standardized tests or by a clear history suggestive of change in functioning over time.

### 3.4. Symptoms Should Not Be Fully Accounted for by Other Disorders

Although comorbid diagnoses of mood or anxiety disorders, substance abuse, or PTSD can be present and do not exclude the diagnosis of TES/CTE, the absence of any other physical or mental condition that could explain the symptoms is mandatory. The differential diagnosis includes a long list of neurodegenerative conditions, infectious diseases, vitamin deficiency syndromes, psychiatric conditions, and physical conditions. 

Vitamin B12 deficiency could lead to sensory and cognitive symptoms, mood disturbance, and motor symptoms, and it is one of the reversible causes of dementia [25]. 

Alzheimer’s disease (AD) should also be considered, especially in patients with executive or memory impairments. AD patients show hippocampal atrophy during brain MRI evaluation, also described in TES/CTE [26]. The difference, however, is made by the history of exposure to repetitive head injuries, the age of symptom onset, the presence of subcortical symptoms, and other manifestations rarely seen in AD. 

Typical cerebrospinal fluid (CSF) changes can also help in difficult cases. Neurosyphilis, known as the great imitator, is also a disease that manifests with motor, cognitive, and behavioural symptoms. However, neurosyphilis can be ruled out by blood and CSF tests [27]. 

Frontotemporal lobar degeneration (FTLD) is a neuropathological condition characterized by symptoms emerging in the presenium, including personality and behavioural changes, and neuropsychological deficits associated with executive dysfunction. Lack of exposure to repetitive head impacts, brain MRI imaging, and thorough consideration of the diagnostic criteria for FTLD could help in the differential diagnosis [28]. 

Multiple system atrophy and corticobasal degeneration are also pathologies manifesting symptoms resembling TES/CTE. In this way, cerebellar symptoms, dysautonomia, and extrapyramidal symptoms, alongside neuroimaging evaluation, as well as lack of exposure to repetitive head impacts and the age of symptom onset, could help in the differential diagnosis [29]. 

It is important to emphasize that the differential diagnosis of the above neurodegenerative conditions could be difficult. Furthermore, exposure to repetitive head impacts, or the history of moderate or severe brain injury, was found to increase the risk of neurodegenerative conditions as well. 

Psychiatric conditions including depression, bipolar disorder, and schizophrenia should also be excluded with the help of a psychiatry expert. Systematic conditions, metabolic disorders, and autoimmune conditions that can affect brain functions also require careful consideration and should be included in the differential diagnosis of TES/CTE.

### 3.5. Staging and Functionality Levels

If all the above steps are satisfied, then TES/CTE can be further determined as per the functional level of the patient: Stage 1, in which the patient does not have any significant changes in their daily activities, job, family, or social roles;Stage 2, when the patient exhibits mild functional limitation with slightly reduced performance in job, household, family, social, or community roles;Stage 3, which is characterized by mild dementia with definite impairment of daily activities;Stage 4, with moderate dementia, where the patient is not independent but can still be taken outside the home for some simple activities;Stage 5, with severe dementia, where the patient cannot participate in any activities outside the home.

The levels of certainty in the diagnosis of TES/CTE (Figure 1) stage the assumed probability of the brain of an individual diagnosed with TES being affected by CTE. However, the group that originally proposed this algorithm referred to it as provisional, suggesting that there are several limitations of available empirical data, mainly originating from American football players (and few other contact sports). Thus, this could indicate that the individuals that are exposed to repetitive head impact events other than those occurring in contact sports could only meet criteria suggestive of CTE [21].

### 3.6. Neuropathological Criteria

Omalu et al. identified p-tau pathology in American football players suffering from CTE in 2005 [4]. CTE pathology has been described in relation to other contact sports and activities including soccer [5], ice hockey [8], wrestling [7], rugby [7,9,10], baseball [7], martial arts, and military service [11,12,16,17,18,19,20,30]; domestic abuse [31]; and single, moderate, or severe TBI [25]. 

McKee et al. proposed 4 pathological stages of CTE based on the severity of p-tau pathology, from stage 1 (mild CTE) to stage 4 (severe CTE) [5]. 

In stage 1, macroscopic examination is unremarkable, while the microscopic examination reveals one or two epicentres of NFTs, astrocytic changes, and dot-like neurites around the small vessels from the depths of the sulci, in the frontal cortex, and less frequently in the temporal, parietal, and insular cortices. Reactive microglia with axonal swelling and distorted profiles can be seen in the subcortical U-fibres. A percentage up to 50% of the brain may show infrequent transactive response DNA binding protein 43 (TDP-43)-positive neurites [5]. 

In stage 2, macroscopic examination shows mild enlargement of the frontal horns of the lateral and third ventricle, cavum septum pellucidum, and discoloration of the locus coeruleus and substantia nigra. Microscopically, there are three or more lesions in multiple cortical areas, larger and superficial in adjacent cortices, and in the locus coeruleus and nucleus basalis of Meynert. Mild TDP-43 pathology and reactive microglia can be seen in clusters in the subcortical U-fibres [5].

In stage 3, there are larger and confluent perivascular patches of NFTs, dot-like and thread-like neurites and astrocytic tangles, and diffuse NFTs in the medial temporal lobe structures including the hippocampus, the entorhinal and perirhinal cortexes, the amygdala, the nucleus basalis of Meynert, and the dorsal and median raphe nuclei. Linear arrays of NFTs and neurites can be seen in the superficial laminae of the cortex. In stage 3, there are also macroscopic features, such as cerebral atrophy, ventricular enlargement, and abnormalities of the septum pellucidum [5]. 

Finally, in stage 4, there is profound atrophy of the cerebrum, the medial temporal lobe, and the diencephalon; depigmentation of the substantia nigra and locus coeruleus; and perivascular p-tau depositions throughout the cerebral cortex. NFTs are also found in the dentate nucleus of the cerebellum, the cerebellar cortex, the spinal cord, and the basis pontis. Extensive astrogliosis and neuronal loss are exhibited with macrovacuolation of layer II in the frontal and temporal lobes [5]. Stage 4 is also characterised by deposits and inclusions that are immunopositive for TDP-43 [5].

The National Institute of Health, with the support of the Foundation for the NIH’s Sports Health Research Program, launched an effort to establish neuropathological criteria for the diagnosis of CTE [32]. 

The first consensus panel evaluated 25 cases of different tauopathies and defined the preliminary diagnostic criteria, which require the presence of phosphorylated tau aggregates with irregular patterns in neurons, astrocytes, and cell processes around small vessels at the depths of the cortical sulci. Additional criteria that needed to be met for a positive diagnosis were the deposition of abnormal tau-immunoreactive pretangles and NFTs, preferably on the superficial layers of the cortex, as well as pretangles and NFTs affecting the CA2 area of the hippocampus and prominent dendritic swelling in CA4, abnormal p-tau immunoreactivity in subcortical nuclei, tau-immunoreactive thorny astrocytes in the subpial and periventricular regions, and tau-immunoreactive large grain-like and dot-like structures [32]. 

The consensus panel met again in 2016 and proposed a working protocol with a minimum threshold for the diagnosis of CTE and an algorithm for the assessment of CTE severity [27]. P-tau aggregates in neurons, with or without thorn-shaped astrocytes, were present at the depth of a cortical sulcus around small blood vessels in the parenchyma and not restricted to the subpial and superficial region of the sulcus, which is the pathognomonic lesion for CTE. In the absence of the pathognomonic lesion and if there is at least clinician concern, tau pathology at the sulcal depth, or superficial cortical NFTs without amyloid-b, then resampling of 4–8 bilateral cortical sulci inducing dorsolateral frontal, orbital frontal, superior middle temporal, and inferior temporal gyri is recommended. If none of the three conditions are met, then the conclusion is not diagnostic for CTE [27]. In the absence of a CTE lesion upon resampling, again, the conclusion will be not diagnostic. If the presence of a CTE lesion is confirmed, then the same procedure as in the case of the pathognomonic lesion’s presence from the beginning should be followed. The conclusion of the neuropathological examination will be highly suggestive of CTE if more than five of the following are present: NFT in the gyral side adjacent to the CTE lesion, NFT in the gyral crest adjacent to the CTE lesion, NFT in superficial cortical layers, NFT in CA4 of the hippocampus, NFT in CA2 of the hippocampus, NFT in the entorhinal cortex, NFT in the amygdala, NFT in the thalamus, NFT in the mamillary body and NFT in the cerebellar dentate nucleus. If less than five of the above are present, then the outcome will be low for CTE [33].

## 4. The Role of Biomarkers

### 4.1. Neuroimaging

#### 4.1.1. Magnetic Resonance Imaging (MRI)

Repetitive brain injuries result in damage to subcortical and cortical microstructures and therefore to changes in white matter integrity visible through diffusion tensor imaging (DTI) MRI [34,35]. Structural changes in the *corpus callosum* [36,37], *cavum septum pellucidum*, and *cavum vergae*, with lower volumes in the amygdala, hippocampus, and cingulate gyrus, are commonly seen on structural brain MRIs of CTE [38,39,40].

#### 4.1.2. Positron Emission Tomography (PET)

Although there are promising developments of tracers such as FDDNP, flortaucipir, and FPT that can detect Tau deposition in CTE, their sensitivity and specificity remain low, and their use is limited [41,42,43,44]. 

#### 4.1.3. Magnetoencephalography (MEG)

Magnetoencephalography is a brain imaging technique with high temporal resolution and has an important role in mild TBI research. It can be used to confirm a positive history of brain dysfunction due to previous brain injuries; however, it cannot be used for the diagnosis of CTE [45].

### 4.2. Fluid Biomarkers

Several studies have shown changes in CSF and blood biomarkers in cases of TBI and repetitive head impacts. Among them are increased serum neurofilament light and heavy polypeptides levels in TBI and repetitive head impact patients [46,47], changes in the neuron-specific enolase levels and serum S-100B calcium-binding protein [48,49,50], increased serum GFAP, IL-6, IL-8, and TNF-a levels [51,52]; and increased plasma and exosomal Tau levels [53,54,55]. YKL-40, amyloid β40 and β42, MCP-4 and MCP-1β, and UCL-L1 have also been studied in mild TBI and repetitive TBI [56,57,58]. Furthermore, more specific biomarkers for brain injury that could offer promising results are the auto-antibodies of glutamate brain receptors that are currently used to evaluate the effects of hypoxia. Despite this evidence on the excitotoxic cascade implications in CTE (being the result of oxidative stress and its effects) and the fact that the interaction between the immunological response to excitotoxic events caused by extracellular glutamate accumulation and calcium imbalance was proposed as a core mechanism of CTE [59,60], little is known about the levels of these auto-antibodies in the brains of CTE patients. 

All the above biomarkers have been extensively studied and can assist with the diagnosis of mild TBI, repetitive head impacts, and PCS. However, from all these biomarkers, the only one that has been studied in a CTE context is the exosomal Tau protein, which was found increased compared to normal controls and seems to be a promising biomarker for the diagnosis of the condition.

### 4.3. Proposed Algorithm for TES/CTE Diagnosis

TES/CTE represents the long-term consequence of repetitive brain injuries or a single moderate or severe brain injury. Although it has gained some publicity in the past few years, and we now understand that it is more common than we thought and has widespread implications for athletes, military personnel, victims of domestic abuse, and patients with epilepsy and repetitive head injuries, its diagnosis remains challenging. In this context, we are proposing a comprehensive algorithm that could allow a more efficient diagnosis of TES/CTE by putting together the previous evidence discussed throughout this study (Figure 2).

## 5. Discussion

The diagnosis of CTE can only be confirmed postmortem, following the recently established neuropathological criteria [32]. There are different proposed diagnostic criteria for the diagnosis of TES. According to the current guidelines in TES/CTE diagnosis and the several mentioned diagnosis criteria [16,17,18,19,20,21], being exposed to unique or repetitive significant head impacts and developing progressive symptoms for at least one year are the main features that suggest a TES/CTE diagnosis and are shared by most of the diagnosis criteria. Recent updates to the diagnosis criteria ought to include additional features that evaluate the cognitive status, as well as the presence of some motor symptoms [20]. Furthermore, the current status of knowledge regarding the TES/CTE diagnosis requires neuropathological confirmation [21]. Despite the definitive character of the neuropathological confirmation, its main limitation is that it can only be performed during postmortem examination. Furthermore, several levels of certainty that vary from low to high depending on the findings are given during this examination. 

In this context, a preventive or at least premortem diagnosis of TES/CTE would be of better use to clinical practice. Although there are numerous studies on neuroimaging, encephalographic, and serum and CSF biomarkers, it seems that this field is not sufficiently mature, and none of them can be currently used for the diagnosis of TES/CTE. 

Promising preliminary results from magnetoencephalography studies and PET scans for the detection of tau deposition, as well as the development of CSF, blood, and salivary biomarkers, may give new insights into a more accurate diagnosis and better understanding of the condition, which then will lead to further studies on treatments and better management of the patients, offering a better quality of life [36,37,38,39,40,41,42,43,44,45]. Prevention, which can be achieved with new TBI protocols in contact sports and for people in high-risk groups, seems to be the golden standard today.

While we did try to minimise some biases in the study design or data analysis, it has to be mentioned that we could not avoid potential biases in the study, such as selection bias, measurement bias, or publication bias. 

## 6. Conclusions

The diagnosis of TES/CTE requires a multidisciplinary approach; thorough investigation for other neurodegenerative disorders, systemic illnesses, and/or psychiatric conditions that can account for the symptoms; and also complex investigations of patient history, psychiatric assessment, and blood and cerebrospinal fluid biomarker evaluation. Thus, we proposed a comprehensive algorithm of diagnosis criteria for TES/CTE based on the similarities and differences between the previous criteria.

## Figures and Tables

**Figure 1 healthcare-11-01689-f001:**
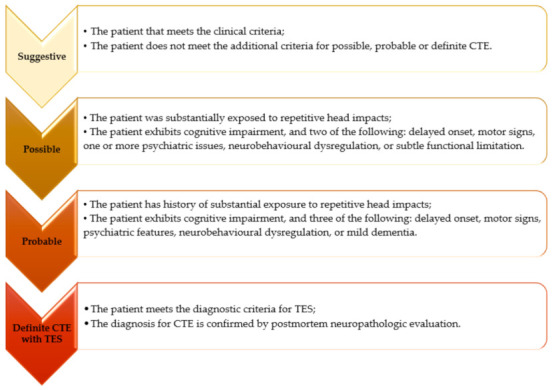
The levels of certainty of CTE when a positive diagnosis for TES is established.

**Figure 2 healthcare-11-01689-f002:**
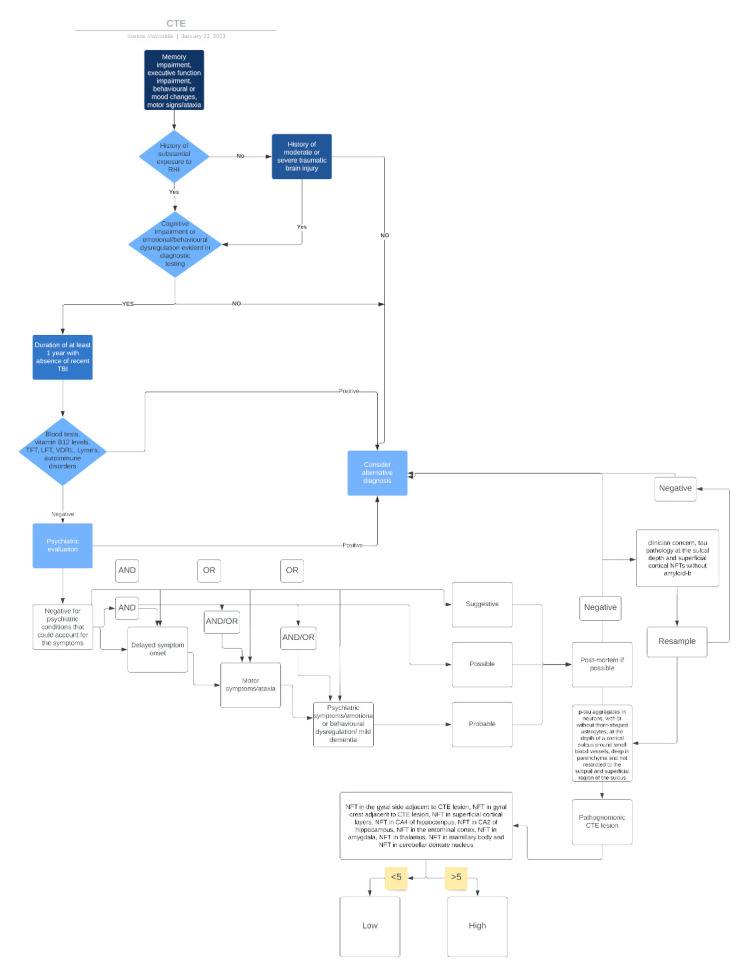
CTE diagnosis algorithm.

**Table 1 healthcare-11-01689-t001:** Current validated TES/CTE diagnostic criteria.

Study	Mandatory Condition	Symptoms	Symptom Duration/ Progression	Comments
Jordan [20]	Exposure to repetitive head impact events	Two or more symptoms: cognitive and/or behavioural impairments; cerebellar dysfunction; pyramidal or extrapyramidal tract symptoms	At least one year, in the absence of new head trauma	Diagnosis can be supported by neuroimaging
Montenigro et al. [19]	Exposure to head trauma events	At least one core symptom: cognitive decline; behavioural impairment; mood disturbance	At least one year; Progressive course, or delayed onset.	Diagnostic criteria should be applied after the exclusion of other conditions that could account for the symptoms.
		Two supportive features: motor signs, impulsivity, anxiety, apathy, paranoia, suicidality		
Reams et al. [16]	Exposure to repetitive head impact events	Cognitive decline, confirmed by neuropsychological testing	Longer than two years; Progressive course, late symptom onset.	Provides classification for TES/CTE according to the predominant symptomatology and based on symptoms progression.
NINDS Consensus [21]	Exposure to repetitive head impact events originating from contact sports, military service, or other causes	Cognitive impairments, episodic memory, and/or executive functioning, and/or neurobehavioural dysregulation.	Progressive course	Progressive course of the symptoms not fully accounted for by any other neurologic, psychiatric, or medical conditions.

## Data Availability

All data is available on request.

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
