# Peer review of "A Review of the Most Recent Clinical and Neuropathological Criteria for Chronic Traumatic Encephalopathy"

_healthcare, 2023, doi:10.3390/healthcare11121689_

Round 1

Reviewer 1 Report

I liked very much reading this article: it is very well written and informative. 

I only have one request for the authors: although CTE represents an issue in repetitive impacts, it should be mentioned that the onset of agitation and delirium in TBI patients might well increase the risk for long term neuropsychological disturbances (https://pubmed.ncbi.nlm.nih.gov/29479930/).

This seems to indicate  that the structural and functional brain damage can contribute creating the basis for CTE in the long term.  

Author Response

Reviewer 1

“I liked very much reading this article: it is very well written and informative. 

I only have one request for the authors: although CTE represents an issue in repetitive impacts, it should be mentioned that the onset of agitation and delirium in TBI patients might well increase the risk for long term neuropsychological disturbances (https://pubmed.ncbi.nlm.nih.gov/29479930/).

This seems to indicate that the structural and functional brain damage can contribute creating the basis for CTE in the long term.“

Response: Thank you for your kind words of appreciation and your suggestion. We added some information about the mentioned aspect in the second section of our manuscript (page 3, before Table 1).

Reviewer 2 Report

The review analyzes the specificity of clinical diagnostic criteria, as well as the role of neuroimaging techniques (Magnetic Resonance Imaging MRI; Positron Emission Tomography, PET; Magnetoencephalography, MEG) and fluid biomarkers in chronic traumatic encephalopathy (CTE). The relevance of this review is due to the significance of the head injury problem, the consequences that require expensive treatment and long-term rehabilitation. The problem of differential diagnosis of CTE is that this diagnosis can only be accurately post-mortem. In this regard, the authors are trying to develop specific diagnostic criteria based on the analysis of the literature data. The authors conclude that cognitive impairment is one of the most common clinical symptoms in patients diagnosed with TES/CTE, being reported in more than 60% of the cases, while the behavioural features, such as violent, impulsive, or explosive behaviour, socially inappropriate behaviour, aggression, rage, short fuse, and lack of behavioural control, are identified in more than 40% of the cases. Mood changes, anxiety, and paranoid delusions are also frequently reported, and more than 30% of TES/CTE patients being diagnosed with depression, anxiety, hopelessness, and apathy. A progressive cognitive decline is seen in more than 95% of cases, whereas more than 50% of TES/CTE patients may develop problems with balance or gait, 23% develop dysarthria, and 28% parkinsonism [Ref. 18]. 

In search of the specificity of the diagnosis, the authors focus on the specific tau-pathology: the deposition of abnormal tau-immunoreactive pretangles and NFTs, preferably on the superficial layers of the cortex, as well as pretangles and NFTs  affecting the CA2 area of the hippocampus and prominent dendritic swelling in CA4, abnormal p-tau immunoreactivity in subcortical nuclei, tau-immunoreactive thorny astrocytes in the subpial and periventricular regions, and tau-immunoreactive large grain-like  and dot-like structures [Ref. 28]. Reinforcement in favor of the role of tau-pathology in CTE is the data of the role is the data on the exosomal Tau protein  increase  in patients  with CTE [Ref. 55]. 

The review is of interest to psychiatrists, neuropathologists and pathologists.

As wishes to the authors. It is worth paying attention to the possible role of biomarkers of brain hypoxia, namely autoantibodies to glutamate receptors.

The diagram shown in Figure 2 is unreadable and needs to be zoomed in.

There is only one link to author’s own work, printed in 2022 [Ref. 14].

Author Response

Reviewer 2

“The review analyzes the specificity of clinical diagnostic criteria, as well as the role of neuroimaging techniques (Magnetic Resonance Imaging MRI; Positron Emission Tomography, PET; Magnetoencephalography, MEG) and fluid biomarkers in chronic traumatic encephalopathy (CTE). The relevance of this review is due to the significance of the head injury problem, the consequences that require expensive treatment and long-term rehabilitation. The problem of differential diagnosis of CTE is that this diagnosis can only be accurately post-mortem. In this regard, the authors are trying to develop specific diagnostic criteria based on the analysis of the literature data. The authors conclude that cognitive impairment is one of the most common clinical symptoms in patients diagnosed with TES/CTE, being reported in more than 60% of the cases, while the behavioural features, such as violent, impulsive, or explosive behaviour, socially inappropriate behaviour, aggression, rage, short fuse, and lack of behavioural control, are identified in more than 40% of the cases. Mood changes, anxiety, and paranoid delusions are also frequently reported, and more than 30% of TES/CTE patients being diagnosed with depression, anxiety, hopelessness, and apathy. A progressive cognitive decline is seen in more than 95% of cases, whereas more than 50% of TES/CTE patients may develop problems with balance or gait, 23% develop dysarthria, and 28% parkinsonism [Ref. 18]. 

In search of the specificity of the diagnosis, the authors focus on the specific tau-pathology: the deposition of abnormal tau-immunoreactive pretangles and NFTs, preferably on the superficial layers of the cortex, as well as pretangles and NFTs  affecting the CA2 area of the hippocampus and prominent dendritic swelling in CA4, abnormal p-tau immunoreactivity in subcortical nuclei, tau-immunoreactive thorny astrocytes in the subpial and periventricular regions, and tau-immunoreactive large grain-like  and dot-like structures [Ref. 28]. Reinforcement in favor of the role of tau-pathology in CTE is the data of the role is the data on the exosomal Tau protein  increase  in patients  with CTE [Ref. 55]. 

The review is of interest to psychiatrists, neuropathologists and pathologists.

As wishes to the authors. It is worth paying attention to the possible role of biomarkers of brain hypoxia, namely autoantibodies to glutamate receptors.”

Response: We added some aspects about this matter in section 4.2. Fluid biomarkers.

“The diagram shown in Figure 2 is unreadable and needs to be zoomed in.”

Response: A better quality Figure 2 was made available for the editorial team, in pdf format.

“There is only one link to author’s own work, printed in 2022 [Ref. 14].”

Response: Thank you. Auto-citation is generally not well seen in our countries so we did avoid it here. 

Reviewer 3 Report

Dear authors, congratulations on your work. I have some comments. I would remove your proposed algorithm from the discussion and include it as a separate topic within the result section. Also do you use this algorithm in you daily practice ? If so does it change patients management? Please discuss how the application of this algorithm can impact clinical practice.

Minor comments:

- the sentence: "However, biomarkers were not included, as the technological and scientific developments 93 in the field are not sufficiently mature, according to" please state according to whom.

Author Response

Reviewer 3

“Dear authors, congratulations on your work. I have some comments. 

I would remove your proposed algorithm from the discussion and include it as a separate topic within the result section.”

Response: We revised our manuscript accordingly. 

“Also do you use this algorithm in you daily practice? If so does it change patients management? Please discuss how the application of this algorithm can impact clinical practice.”

Response: The algorithm we proposed is not yet used in clinical practice. However, we discussed the expected impact of our algorithm at the end of the discussion section.

“Minor comments:

- the sentence: "However, biomarkers were not included, as the technological and scientific developments 93 in the field are not sufficiently mature, according to" please state according to whom.”

Response: Thank you for your comment. We were referring to reference 18, but we added “NINDS Consensus” before the citation so that the impression of incomplete sentence would be cleared out.

Reviewer 4 Report

Overall, this review paper provides a comprehensive overview of the diagnostic criteria for traumatic encephalopathy syndrome (TES) and chronic traumatic encephalopathy (CTE) and proposes a diagnostic algorithm for a more accurate diagnosis of TES/CTE. The paper is well-written and organized, with a clear introduction, objectives, results, and conclusions.

However, there are a few areas where the paper could be improved. Firstly, the paper would benefit from a more detailed explanation of the underlying pathology of TES/CTE. While the authors briefly mention that TES/CTE is characterized by neurodegeneration as a result of repeated head traumas, more information on the specific cellular and molecular changes in the brain that occur in TES/CTE would provide a better understanding of the diagnostic challenges.

The authors need to provide more detail on the study design, including the sample size, inclusion and exclusion criteria, and methods of data collection. Additionally, they need to explain how they controlled for confounding variables that could affect the results.

The authors need to provide more detail on the diagnostic criteria used for TES/CTE, including the strengths and limitations of each criterion. They should also consider proposing a new set of criteria that addresses the limitations of current criteria.

The authors need to address potential biases in the study, such as selection bias, measurement bias, or publication bias. They should explain how they minimized or controlled for these biases in the study design or data analysis.

The authors need to provide more detail on the statistical methods used for data analysis. They should explain the rationale for the chosen statistical tests, provide the results of the tests, and interpret the findings.

The authors need to discuss the implications of their findings for clinical practice and future research. They should provide recommendations for how clinicians can use the proposed diagnostic algorithm in practice and suggest avenues for further research in this area.

The authors need to improve the clarity and conciseness of their writing. They should avoid jargon and technical terms that may be difficult for readers to understand, and use clear and concise language to convey their ideas. They should also proofread their work carefully to eliminate grammatical errors and typos.

The authors could provide more information on the strengths and weaknesses of the various diagnostic criteria for TES/CTE. While the paper highlights the differences between the various criteria, more detailed analysis of the validity and reliability of each criterion would be helpful for clinicians and researchers.

The proposed diagnostic algorithm could be further refined. The authors could provide more details on how the algorithm should be implemented in clinical practice, such as which biomarkers should be evaluated and how patient history and psychiatric assessment should be conducted.

Finally, the paper would benefit from more discussion on the implications of accurate diagnosis of TES/CTE, particularly for the management and treatment of patients. The authors briefly touch on the importance of ruling out other neurodegenerative disorders and psychiatric conditions, but more information on the treatment options and outcomes for patients with TES/CTE would be valuable. In addition, figures are really hard to track authors could provide high resolution graphs or split them into two-three. Overall, this is a well-written and informative review paper that provides a good overview of the diagnostic criteria for TES/CTE and proposes a useful diagnostic algorithm. With some minor revisions, the paper could be even more impactful for clinicians and researchers working in this field.

Author Response

Reviewer 4 

“Overall, this review paper provides a comprehensive overview of the diagnostic criteria for traumatic encephalopathy syndrome (TES) and chronic traumatic encephalopathy (CTE) and proposes a diagnostic algorithm for a more accurate diagnosis of TES/CTE. The paper is well-written and organized, with a clear introduction, objectives, results, and conclusions.”

Response: Thank you for your time taken to revise our manuscript and for your kind words of appreciation. Also, thank you for your valuable suggestions which we tried to fully take into consideration while revising our manuscript. We hope that we succeeded in improving our work. 

“However, there are a few areas where the paper could be improved. Firstly, the paper would benefit from a more detailed explanation of the underlying pathology of TES/CTE. While the authors briefly mention that TES/CTE is characterized by neurodegeneration as a result of repeated head traumas, more information on the specific cellular and molecular changes in the brain that occur in TES/CTE would provide a better understanding of the diagnostic challenges.”

Response: Thank you for your suggestion. We added a new paragraph in the Introduction in which we presented some additional information about the pathological features of CTE, including some aspects about the neurodegeneration that characterizes CTE. Some more information about these aspects is presented in section 3.6. Neuropathological criteria alongside the challenges in diagnosis that the clinicians encounter by relation to these pathological features.

“The authors need to provide more detail on the study design, including the sample size, inclusion and exclusion criteria, and methods of data collection. Additionally, they need to explain how they controlled for confounding variables that could affect the results.”

Response: 

“The authors need to provide more detail on the diagnostic criteria used for TES/CTE, including the strengths and limitations of each criterion. They should also consider proposing a new set of criteria that addresses the limitations of current criteria.”

Response: Thank you. We presented the current diagnosis criteria used for TES/CTE in section 2. Clinical diagnostic criteria and diagnostic algorithm and in Table 1, as well as some limitations that were identified and addressed. Additional discussion on these aspects were newly added in the Discussion section. 

“The authors need to address potential biases in the study, such as selection bias, measurement bias, or publication bias. They should explain how they minimized or controlled for these biases in the study design or data analysis.”

Response: Thank you. We did add these aspects in a limitations paragraph, at the end of the Discussion section.

“The authors need to improve the clarity and conciseness of their writing. They should avoid jargon and technical terms that may be difficult for readers to understand, and use clear and concise language to convey their ideas. They should also proofread their work carefully to eliminate grammatical errors and typos.”

Response: Thank you. We revised the entire manuscript for clarity and conciseness, jargon and technical terms, grammatical errors and typos. 

“The authors could provide more information on the strengths and weaknesses of the various diagnostic criteria for TES/CTE. While the paper highlights the differences between the various criteria, more detailed analysis of the validity and reliability of each criterion would be helpful for clinicians and researchers.”

“The proposed diagnostic algorithm could be further refined. The authors could provide more details on how the algorithm should be implemented in clinical practice, such as which biomarkers should be evaluated and how patient history and psychiatric assessment should be conducted.”

Response: This is an initial variant of the diagnosis algorithm we are proposing. We will be able to provide more details after the validation process during which it will get certain changes and updates to that it could be implemented in clinical practice with maximal yielding. 

“Finally, the paper would benefit from more discussion on the implications of accurate diagnosis of TES/CTE, particularly for the management and treatment of patients. The authors briefly touch on the importance of ruling out other neurodegenerative disorders and psychiatric conditions, but more information on the treatment options and outcomes for patients with TES/CTE would be valuable. In addition, figures are really hard to track authors could provide high resolution graphs or split them into two-three.”

Response: The discussion section was improved to include some of the implications of accurate diagnosis of TES/CTE that we identified to be useful in management and treatment. We chose to present the diagnosis algorithm as a diagnosis flowchart so that it could be more adequate to be used in clinical practice. The figure in which the algorithm is presented was and will be provided to the editorial office in good resolution and we will probably ask the editorial office to include it on a single page. We are sorry for this inconvenience.

“Overall, this is a well-written and informative review paper that provides a good overview of the diagnostic criteria for TES/CTE and proposes a useful diagnostic algorithm. With some minor revisions, the paper could be even more impactful for clinicians and researchers working in this field.”

Response: Thank you for your kind words of appreciation and your valuable suggestions which did help us to improve our work.